



# Inter-Comparison of Thermal-Optical Carbon Measurements by Sunset and DRI Analyzers Using the IMPROVE_A Protocol

Xiaolu Zhang, Krystyna Trzepla, Warren White, Sean Raffuse, Nicole Pauly Hyslop

Air Quality Research Center, University of California, Davis, CA

## Abstract

Thermal-Optical Analysis (TOA) is a class of methods widely used by long-term air quality monitoring networks for determining organic carbon (OC) and elemental carbon (EC) in atmospheric aerosols collected on filters. Results from TOA vary not only with differences in
operating protocols for the analysis, but also with details of the instrumentation with which a given protocol is carried out. Three models of TOA carbon analyzers have been used for the IMPROVE_A protocol in the past decade within the Chemical Speciation Network (CSN). This study presents results from inter-comparisons of these three analyzer models using two sets of CSN quartz filter samples, all analyzed with the IMPROVE_A protocol. One comparison was
between the Sunset Model 5L (Sunset) analyzers and the Desert Research Institute (DRI) Model 2015 (DRI-2015) analyzers, using 4073 CSN samples collected in 2017. The other comparison was between the Sunset and the DRI Model 2001 (DRI-2001) analyzers, using 303 CSN samples collected in 2007.

Both comparisons showed a high degree of inter-model consistency in total carbon (TC) and the
major carbon fractions, OC and EC, with mean bias within 5% for TC and OC, and within 12% for EC. Relatively larger and diverse inter-model differences (mean biases of 5% – 140%) were found for thermal subfractions of OC and EC (i.e. OC1-OC4 and EC1-EC3), with better agreement observed for subfractions with higher mass loadings and smaller within-model uncertainties. Optical charring correction was found to be critical in bringing OC and EC
measurements by different TOA analyzer models into better agreement. Appreciable inter-model differences in EC between Sunset and DRI-2015 (mean bias ± SD of 21.7% ± 12.2%) remained for ~ 5% of the 2017 CSN samples; examination of these analysis thermograms revealed that the optical measurement (i.e. filter reflectance and transmittance) saturated in the presence of strong absorbing materials on the filter (e.g. EC), with excessive absorption leaving insufficient
dynamic range for detection of carbon pyrolysis, thus no optical charring correction. Differences in possible instrument parameters and configuration related to disagreement in OC and EC subfractions are also discussed.

Our results provide a basis for future studies of uncertainties associated with the TOA analyzer model transition in assessing long-term trends of CSN carbon data. Further investigations using
these data are warranted focusing on the demonstrated inter-model differences in OC and EC subfractions. The within- and inter- model uncertainties are useful for model performance evaluation.



## 1. Introduction

Carbonaceous aerosols are a major component of ambient $PM_{2.5}$ (Zhang et al., 2007), which has
important effects on visibility (Watson, 2002), health (Pope and Dockery, 2006) and regional to
global climate (IPCC, 2007). Thermal-Optical Analysis (TOA) is a conventional method
employed by long-term monitoring networks to distinguish organic carbon (OC) from elemental
carbon (EC) in quartz filter samples of $PM_{2.5}$. In this method, carbonaceous aerosols are
separated into OC and EC by recording carbon evolved under programmed progressive heating,
initially in an inert atmosphere, followed by further heating with oxygen present, after making an
optically-guided correction for the effects of sample charring (Huntzicker et al., 1982). The
resulting OC-EC split is sensitive to details of the heating sequence and atmosphere, as well as
the optical correction procedure.

The Chemical Speciation Network (CSN) was created to support implementation of the 1997
$PM_{2.5}$ National Ambient Air Quality Standards (NAAQS) (EPA, 1997). Within the network, 24-
hr $PM_{2.5}$ samples are collected on different filter media (e.g. PTFE, Nylon, and Quartz) at
approximately 160 sites across the U.S., most of which are located in urban areas, and are
analyzed for $PM_{2.5}$ chemical components. Since inception, CSN has been using the TOA method
for carbon analysis on quartz filters but with different sample-collection methods, thermal-
optical analytical protocols and instrumentation (Spada and Hyslop, 2018). Prior to 2007, CSN
used varied sampler designs for collecting carbon samples on 47 mm diameter quartz filters,
from which OC and EC were determined by Sunset analyzers that implemented NIOSH
thermal/optical transmittance (TOT) protocol (Birch and Cary, 1996). During the years 2007 -
2009, the network transitioned to using the URG-3000N samplers to collect carbon samples on
25 mm diameter quartz filters, coinciding with the change in the analytical protocol from NIOSH
TOT to IMPROVE_A thermal/optical reflectance (TOR) (Chow et al., 2007), to be more
consistent with the U.S. Interagency Monitoring of PROtected Visual Environments
(IMPROVE) network. Since late 2009, no change has occurred in the analytical protocol or
sample collection, but there were two TOA analyzer model transitions. As shown in Figure 1, in
the beginning of 2016, TOA carbon analysis for CSN transitioned from using the Desert
Research Institute (DRI) Model 2001 analyzers (termed "DRI-2001" hereinafter) to DRI Model
2015 multi-wavelength analyzers (termed "DRI-2015" hereinafter) , and again in October 2018,
CSN TOA transitioned from using DRI-2015 analyzers to Sunset Laboratory Model 5L
analyzers (termed "Sunset" hereinafter). In addition to the abovementioned changes, the network
started blank subtraction on carbon data in November 2015.

While measurement differences among thermal protocols (e.g., IMPROVE_A, NIOSH,
EUSAAR, etc.) and between optical corrections (e.g. reflectance vs. transmittance) have been
extensively studied and documented in the literature (e.g., Conny et al., 2003; Chow et al., 2004;
Watson et al., 2005; Khan et al., 2012; Chan et al., 2019), less attention has so far been given to
possible differences in OC-EC splits produced by nominally identical analytical protocols carried
out on differently designed and manufactured instrument systems. Most inter-model comparisons
were focused on examining variations between different units of the same model (e.g. Schauer et
al., 2003; Ammerlaan et al., 2015). A previous study by Chow et al. (2015) compared results
from the 2001 and 2015 models of the DRI analyzers using 67 urban (from Fresno Supersite)
samples and 73 rural (from IMPROVE network) samples and concluded that no significant
difference was found in EC or OC reported by the two models. Wu et al. (2012) compared a
Sunset analyzer and a DRI-2001 analyzer using ~100 ambient samples collected in the Pearl



River Delta in China and reported similar consistency for OC and EC. While these studies provided insights on the inter-model comparisons of different TOA analyzers, their sample sizes were limited.

The goal of this study is to characterize the consistency and differences in the results reported from the three TOA models successively deployed in the CSN network to run the same protocol, IMPROVE_A. Two models, the DRI-2001 manufactured by Atmoslytic, Inc and the DRI-2015 manufactured by Magee Scientific, were designed by DRI specifically to carry out versions of the IMPROVE protocol. The third model, the Sunset Model 5L designed and manufactured by Sunset Laboratory, Inc, is marketed to carry out both the NIOSH and IMPROVE_A protocols. For each model type there have been multiple units dedicated for CSN carbon analysis in the past decade, including eight DRI-2001units (Chow et al., 2007), 13 DRI-2015 units, and five Sunset units. Two sets of 25 mm diameter quartz filter samples from the CSN network were analyzed, each by a pair of models, for TC, OC, EC and thermal subfractions (OC1-OC4, EC1-EC3, and OP). Findings from these two comparisons provide a basis for accounting for TOA model transitions in future studies of CSN carbon long-term trends. Information such as within- and inter- model uncertainties between Sunset and DRI analyzers is also useful for studies evaluating model predictions against CSN data (e.g., Emery et al., 2017), as well as source apportionment studies using speciated $PM_{2.5}$ carbon data (e.g., Kim and Hopke, 2005; Liu et al., 2006).

## 2. Methods
### 2.1 Instrumentation in Comparison

Table 1 lists the major differences among the three TOA carbon analyzer models used in the inter-comparisons. The differences in laser source, detector and temperature calibration method are discussed in more details as follows.

*Laser Source*: The Sunset and DRI-2001 analyzers employ a single wavelength laser source for measurement of filter reflectance and transmittance. The Sunset analyzer uses a diode laser at 658 nm, whereas DRI-2001 employs a Helium-neon (He-Ne) laser at 633 nm. DRI-2015 employs seven diode lasers with differing wavelengths from 405 nm to 980 nm (Chen et al., 2015). For CSN samples analyzed by DRI-2015, the 635 nm EC data, reported as EC by reflectance, is considered equivalent to the 633 nm data reported by the DRI-2001 analyzer (Chen et al., 2015) and is therefore used in this study for comparison with the Sunset measurements.

*Detector*: Both DRI-2001 and Sunset analyzers use a flame ionization detector (FID) that quantifies $CH_4$, whereas DRI-2015 uses a non-dispersive infrared (NDIR) detector to quantify $CO_2$. These two types of detectors have distinctly different response to interference and noise levels, thus different signal integration methods are used (further discussed in Sect. 3.2.2).

*Temperature calibration*: Temperature calibration in TOA refers to the method used to adjust oven temperature based on the response of an external temperature-indicating device. Sunset and DRI analyzers adopt fundamentally different methods to calibrate the temperature plateaus in the IMPROVE_A protocol. In a Sunset analyzer, a thermocouple, positioned ~2 cm downstream of the sample filter holder, is used to monitor sample temperature at each IMPROVE_A temperature set point during an analysis. The distance between the thermocouple and sample punch is accounted for in temperature calibration by placing another thermocouple at the sample

punch position, measuring the difference between the readings from the two probes, and adjusting the settings in the thermal analytical protocol accordingly (i.e. temperature offsets). The temperature offsets in Sunset analyzers can vary greatly per temperature step depending on the heat dissipation inside the oven (Panteliadis et al., 2015; Phuah et al., 2009). On the other

hand, DRI used Tempilaq° G, a type of quick-drying chemical, as temperature indicators in the temperature calibration for both analyzer models (DRI Standard Operating Procedure, 2016; Chow et al., 2005). Briefly, six Tempilaq° G liquids that change optical properties at 121, 184, 253, 510, 704, and 816°C were used in calibrating the six IMPROVE_A temperature plateaus (140, 280, 480, 580, 740 and 840°C). During the analysis of each Tempilaq° G sample, the oven

temperature is slowly incremented to a narrow range near the temperature where the specific Tempilaq° G changes color, while the laser reflectance and transmittance are monitored for a sharp rise in response to the change. The sample oven temperature values are regressed on the corresponding Tempilaq° G temperatures and are interpolated/extrapolated to the IMPROVE_A temperatures based on the linear regression slope and intercept.

**2.2 Experimental Data**

### 2.2.1  Thermal-Optical Analysis

For TOA with the IMPROVE_A analytical protocol, a punch of approximately 0.5-0.6 cm$^2$ in size was taken from each filter sample using a precision tool and inserted into the sample oven.

Owing to the destructive nature of the TOA method and the limited sample deposit area of the 25 mm diameter quartz filter (3.53 cm$^2$), only a maximum of three 0.5-0.6 cm$^2$ circular punches can be taken from one filter sample. The filter punch was first heated in an inert (100% He) atmosphere where various OC subfractions volatilized at 140°C (OC1), 280°C (OC2), 480°C (OC3) and 580°C (OC4), respectively. The system was then switched to an oxidizing atmosphere

(He with a fixed amount of O$_2$) where EC subfractions combusted at 580°C (EC1), 740°C (EC2) and 840°C (EC3). The liberated carbon compounds are converted to either carbon dioxide (CO$_2$) or methane (CH$_4$), followed by infrared absorption (CO$_2$) or flame ionization (CH$_4$) detection.

During the thermal analysis, a fraction of the OC pyrolyzes or chars under the inert He atmosphere into EC-like substances. The formation of pyrolyzed OC (OP) can bias the

estimation of EC high and OC low. To correct for this interference, the reflectance and transmittance of the sample filter were monitored throughout the analysis using a laser source. The filter reflectance and transmittance decreased in response to the formation of OP and then increased as the OP was combusted. The reflectance or transmittance split between OC and EC is defined as the point when reflectance or transmittance returned to its initial reading before the

heating started.

### 2.2.2  Sample Description

Two sets of CSN carbon samples collected on 25 mm diameter quartz filters were analyzed respectively in the two inter-model comparisons. Set 1 consists of 303 CSN filters sampled in

September and October of 2007 that were previously analyzed by DRI with the DRI-2001models in the year of 2008 (Figure 1). These filters were retrieved from cold storage and re-analyzed by UC Davis using Sunset analyzers in 2017-2018. No blank or replicate measurements were available in this set due to sample unavailability. Set 2 consists of 4073 CSN samples and 622 CSN field blanks collected between May and September of 2017, which were sequentially





analyzed by the Sunset analyzers at UC Davis and by the DRI-2015 analyzers at DRI within a
     year after sample collection. Replicate analyses were performed using both analyzer models
     within this set and were used to evaluate the within-model uncertainty (detailed in Sect. 2.2.3).

     Sunset raw data were processed using a custom R computing package developed by UC Davis
     (hereinafter referred to as "UCD-Sunset data processing"). The program slightly modifies the
algorithms provided by the Sunset calculation software (Version 423) in that 1) premature EC
     evolution was not considered and 2) no correction was made for the dependency of laser
     reflectance on temperature. All Sunset and DRI-2015 data are reported in mass loadings (in
     $\mu g/cm^2$). DRI-2001 data from the archived 2007 CSN samples were downloaded from the EPA
     Air Quality System (AQS) database (https://aqs.epa.gov/api/rawData). The concentration data (in
$\mu g/m^3$) were converted to mass loadings (in $\mu g/cm^2$) using nominal sample volume (33 m$^3$) and
     filter area (3.53 cm$^2$) for direct comparison against the Sunset data. Unless otherwise noted, the
     OC, EC and OP data discussed below refer to those determined by the reflectance optical
     correction.

**2.2.3  Quality Control**

     - *Blank measurement*
       Table 2 summarizes the mean and standard deviation of the carbon mass loadings from
       measurements of 622 CSN field blanks by the Sunset and DRI-2015 analyzers. OC and EC
levels on blank filters are minimal. The difference between analyzers is also trivial. Sunset
       and DRI-2015 mass loading data were not blank subtracted to allow for direct comparison
       with the DRI-2001 data.

     - *Calibration*
The FID and NDIR detector responses are normalized to a known amount of CH$_4$ gas (i.e.
       5% CH$_4$ in Helium gas by mixing ratio) that is injected at the end of each sample analysis. In
       addition, the detector linearity was verified and calibrated by a set of carbon-containing
       aqueous solutions. Specifically, sucrose (C$_{12}$H$_{22}$O$_{11}$) standards with concentration spanning
       from 2 to 210 $\mu g$ Carbon /cm$^2$ were used to calibrate the Sunset analyzers (UCD, 2019). The
two DRI models were calibrated using 5 to 20 $\mu l$ of 1800 ppm Sucrose and KHP (C$_8$H$_5$KO$_4$)
       solutions (DRI, 2012, 2016). The split between OC and EC cannot be calibrated or verified
       due to the lack of reference material for EC (Baumgardner et al., 2012).

     - *Measurement Uncertainty*
Measurement uncertainty of the Sunset and DRI-2015 analyzers were estimated separately
       utilizing data from replicate analyses (i.e. two analyses on the same filter sample by the same
       analyzer model). Within the 2017 sample set, a total of 519 and 518 samples were replicated
       by Sunset and DRI-2015 analyzers, respectively. Due to the limited sample deposit area of
       CSN 25 mm diameter quartz filters, the replicate analyses by Sunset and DRI analyzers were
performed on different filters.

     The scaled relative difference (SRD) for each sample is calculated using Eq. 1, where
     [Original]$_i$ and [Replicate]$_i$ represent the mass loadings of the original and replicate paired
     analyses on the same filter.


$$SRD_i = \frac{([Original]_i - [Replicate]_i)/\sqrt{2}}{([Original]_i + [Replicate]_i)/2} \times 100 \qquad (1)$$

The SRD, which equals relative difference (RD) divided by $\sqrt{2}$, is chosen over RD because it accounts for the presence of equal errors in both original and replicate measurements (Hyslop and White, 2009). The mean value $\overline{SRD}$ provides an estimate of the within-model replication bias, which was negligible, and the standard deviation (1σ) of SRD provides an estimate for the within-model measurement uncertainty ($Unc$).

$$Unc = \left(\frac{1}{n}\sum(SRD_i - \overline{SRD})^2\right)^{1/2} \qquad (2)$$

Figure 2 illustrates the relationship between SRD and mass loading for TC, OC and EC measured by Sunset (a-c) and DRI-2015 (d-f). As expected, the within-model replication bias is close to zero for both Sunset and DRI-2015 because the replicate and original analysis are essentially identical. For all three components, and particularly for EC where some measurements are near the method detection limit (MDL) (illustrated by the vertical dashed line at 0.2 µg/cm$^2$ in the plots), SRD decreases with increasing mass loading. While all analysis pairs are included in Figure 2, those with a mean mass loading less than 3 times the MDL are excluded from the calculations of $Unc$ to obtain a stable estimate of measurement uncertainty.

Assuming the within-model uncertainties are independent, the combined inter-model uncertainty ($Unc_{inter}$) can be calculated by Eq. 2a, where $Unc_{DRI}$ and $Unc_{Sunset}$ are the within-model uncertainties determined for DRI-2015 and Sunset analyzers using replicate analyses, respectively.

$$Unc_{inter} = \sqrt{(Unc_{DRI})^2 + (Unc_{Sunset})^2} \qquad (2a)$$

The overall measurement bias and uncertainty for all carbon components are summarized in Table 3, which provide benchmarks for inter-model comparison discussed in the following sections. For most components, uncertainties estimated for the Sunset and DRI-2015 analyzers were comparable, except for OP and OC1, where DRI-2015 uncertainties were a factor of 2-4 larger.

## 3. Results and Discussion
### 3.1 Inter-Model Comparison of Carbon Measurements

This section presents results from the two inter-model comparisons for bulk TC, OC, and EC, as well as for individual thermal subfractions (OC1, OC2, OC3, OC4, OP, EC1, EC2, and EC3). Arithmetic differences (AD) (Eq. 3) and scaled relative differences (SRD) (Eq. 4) are calculated between results from Sunset and DRI-2001 analyzers using the 2007 CSN sample set, as well as between results from Sunset and DRI-2015 using the 2017 CSN sample set. In calculating SRD, the underlying assumption is that the observed differences are equally allocated to measurements from the two models in comparison; because no standard reference materials are available for the TOA measurement technique, there is no way to allocate the errors to a particular laboratory or analyzer model.

$$AD_i = [Sunset]_i - [DRI]_i \qquad (3)$$



$$SRD_i = \frac{([Sunset]_i - [DRI]_i)/\sqrt{2}}{([Sunset]_i + [DRI]_i)/2} \times 100 \qquad (4)$$

255 Figure 3 shows the probability density curves of SRDs for Sunset vs. DRI-2001 (purple), and Sunset vs. DRI-2015 (orange). The location of the peak relative to the x-axis center (or as measured by mean of the SRDs) indicates systematic inter-model bias that occurred for the majority of the data points, while the spread of the curve (or as measured by standard deviation of the SRDs) represents variability/coherence of these biases. Also shown in Figure 3 are the 260 within-model uncertainties determined from replicate analyses (Table 3), albeit only available for Sunset vs. DRI-2015, to assist in the interpretation of the inter-model SRDs. $R^2$ values are tabulated as an indicator of the degree of linear correlations between the two models. The means and standard deviations of ADs and SRDs are summarized in Table 4.

### 3.1.1 Bulk TC, OC and EC

265 TC and the major carbon fractions, OC and EC, exhibited good agreement in both comparisons, with the smallest SRDs and highest $R^2$ values found for TC (SRDs = $-1.6 \pm 5.4\%$ and $R^2 = 0.98$ for Sunset vs DRI-2001, and SRDs = $-0.9 \pm 6.0\%$ and $R^2 = 0.99$ for Sunset vs. DRI-2015). Between Sunset and DRI-2015, the ADs of TC (e.g., $-0.5 \pm 2.0\,\mu g/cm^2$) were comparable to the difference in TC measured from the blank filters (Table 2). The consistency in the TC 270 measurements over a wide temporal range, indicated by the similar TC mass loadings from the original analysis by DRI-2001 and the reanalysis by Sunset 10 years after sample collection, suggests good measurement reproducibility for TC as well as sample stability in long-term cold storage for bulk carbon fractions.

Relative to TC, similar but slightly weaker inter-model correlations were found for OC ($R^2 =$ 275 0.95 for Sunset vs DRI-2001and 0.98 for Sunset vs. DRI-2015) and EC ($R^2 = 0.95$ for Sunset vs DRI-2001 and 0.90 for Sunset vs. DRI-2015) (Figure 3b and 3c). Sunset OC was lower than those determined by the two DRI analyzers by similar amount, with an average AD of ~1.5 $\mu g/cm^2$ and SRD of ~4% (Table 4). Sunset EC was higher when compared to the two DRI analyzers, and the inter-model difference varied by a factor of two in terms of SRD ($6.5 \pm 8.3\ \%$ 280 and $11 \pm 15\ \%$ relative to DRI-2001 and DRI-2015, respectively). Mean SRDs, or inter-model bias, of all three carbon components did not exceed the combined inter-model uncertainties for Sunset vs. DRI-2015; the mean SRD of EC (11%) was the largest and closest to its inter-model uncertainty (12%), suggesting the results are not statistically different. The consistently opposite inter-model biases of OC and EC from the two pairs of comparisons suggested disagreement in 285 the OC-EC split by Sunset and DRI analyzers.

### 3.1.2 Thermal OC and EC subfractions

An examination of individual thermal OC and EC subfractions revealed large and diverse inter-model differences in these subfractions, a phenomenon referred to as "carbon migration" by some previous studies (e.g., Chow et al., 2007). In general, subfractions with higher mass 290 loadings (e.g., OC2, OC3 and EC1) showed better inter-model agreement, with mean SRDs within ~20% and $R^2$ above ~0.8 (Figure 3); these subfractions also had smaller within-model uncertainties (Table 3). Relatively larger inter-model SRDs were observed for OC1, OC4, EC2 and EC3, coinciding with their lower mass loadings. EC3, the smallest subfraction in terms of mass loading (Table 4), showed the lowest degree of inter-model agreement among all OC and 295 EC subfractions. DRI analyzers reported many more EC3 data points below the MDL than Sunset, leading to some SRD values far beyond 100% (Figure 3k). The most volatile subfraction,



OC1, exhibited the largest inter-model SRDs among all four OC subfractions. Evaporative loss during handling and storage of the samples could artificially reduce the mass loading of OC1. Although good sample stability was demonstrated`1 1 for bulk TC, it is possible that the 82% bias of Sunset OC1 relative to DRI-2001 was primarily due to evaporation of OC1 during long-term storage.


Systematic inter-model biases (as measured by the mean SRDs) diverged in terms of both magnitude and direction across different thermal subfractions. Relative to DRI analyzers, Sunset measured lower OC1, OC3, and OC4, and higher OC2. Despite the small average mass loadings of OC1 and OC4, they showed much higher ADs than OC2 and OC3 (Table 4). In contrast to OC, the two DRI analyzers measured all three EC subfractions lower than Sunset. The degree of inter-model differences varied greatly with subfraction and model pair, from 5.4% for EC1 between Sunset and DRI-2001 up to 137% for EC3 between Sunset and DRI-2015.


Collectively, inter-model SRDs of the summed OC subfraction mass loadings (OC1 + OC2 + OC3 + OC4 = $OC_{sum}$) were –14% and –16% when Sunset was compared to DRI-2001 and DRI-2015, respectively, substantially larger than the differences in OC after charring correction (–4.6% and –4.1%). The summed EC subfraction mass loadings (EC1+EC2+EC3 = $EC_{sum}$) differed by 14% and 29% for Sunset vs. DRI-2001 and Sunset vs. DRI-2015, respectively, also much higher than those in the optically-corrected EC (6.5% and 11%).


OP is a thermal fraction formed as a result of OC pyrolysis, which is strongly dependent on thermal parameters and instrument configuration (Cavalli et al., 2010, Yu et al., 2002, Zhi et al., 2009). In our results, Sunset OP was on average 38% and 66% higher than DRI-2001 and DRI-2015, respectively. Arithmetically, the inter-model differences in terms of absolute mass loadings of OP (ADs = 2.1 and 2.9 µg/cm$^2$) corresponded to a large fraction of the observed differences in $OC_{sum}$ (56% and 67%) and $EC_{sum}$ (75% and 76%) for both model pairs (Sunset vs. DRI-2001 and Sunset vs. DRI-2015, respectively). Optical charring correction reduced the inter-model biases in OC and EC relative to those of $OC_{sum}$ and $EC_{sum}$, discussed in detail below.



## 3.2 Understanding inter-model differences in TOA results

In this section, we further investigate the causes of the inter-model differences, with a focus on the role of optical charring correction in the final OC-EC split, as well as the instrument differences that are possibly related to the observed migration among OC and EC subfractions.

### 3.2.1 Optical charring correction

Optical correction is an essential component of the TOA method to remove measurement artifacts in OC and EC caused by charring. Specifically, OC, EC, and their thermal subfractions without and with optical charring correction are related as follows:


$$OC_{sum} \text{ (uncorrected OC)} = OC1+OC2+OC3+OC4 \quad (5)$$

$$EC_{sum} \text{ (uncorrected EC)} = EC1 + EC2 + EC3 \quad (6)$$

$$OC \text{ (corrected OC)} = OC_{sum} + OP \quad (7)$$

$$EC \text{ (corrected EC)} = EC_{sum} - OP \quad (8)$$





Equations 5-8 show that, without correction, OP, the charred fraction of OC, would be reported as part of EC, leading to an overestimate of EC and an underestimate of OC by the same amount that equals the mass of OP.

Shown in Figure 4 are box-whisker plots of SRDs in uncorrected and corrected OC and EC between Sunset and DRI-2015, grouped by 20 equal-sized percentile bins (5%) of their average mass loadings. Optical charring correction brought results into better agreement with reduced SRDs across their whole range of mass loadings for both OC and EC, which is not surprising given the large ADs in OP that were equivalent to 67% and 76% of ADs of $OC_{sum}$ and $EC_{sum}$, respectively. The remaining inter-model differences in EC, larger than those of OC, and the

varying EC SRDs across its mass loading range are worth noting. In particular, the highest (95th percentile and above) EC mass loadings had a median SRD of 19%, far exceeding the median SRDs in the lower mass loading percentiles ($\leq$ 12%). In investigating this anomaly, we found that EC SRDs were larger for samples with no instrumentally detected OP (i.e. OP = 0) by both Sunset and DRI-2015 analyzers, as shown in Figure 5a. Figure 5b further revealed that

approximately 30% of the samples in the highest EC mass loading bin have OP equaling 0, meaning no optical charring correction on the final reported mass loadings of EC or OC for these samples. In total, out of the 4073 CSN samples analyzed by Sunset and DRI-2015, 179 samples had no charring correction determined by both analyzers, with an additional 324 samples having no charring correction determined by only the DRI-2015 analyzers. As shown in Figure 5c, for

the 179 samples with no correction from both Sunset and DRI-2015, considerable correlation was found between inter-model differences of EC and $OC_{sum}$, reflecting that EC biases almost solely originated from the thermal effect, as a result of the absence of measured OP and thus lack of charring correction. Samples with charring correction (i.e. OP > 0) showed essentially no correlation between the inter-model biases of EC and $OC_{sum}$.

The prevalence of CSN samples with no instrumentally detected OP, especially samples with high EC loadings (Figure 5b), is intriguing and was investigated by a close examination of thermograms of all the 2017 CSN samples analyzed by Sunset. Figure 6a illustrates typical laser transmittance profiles from a Sunset analyzer for a sample with no charring correction (i.e. OP = 0), a normal sample with charring correction (i.e. OP > 0), along with a blank sample; the

transmittance profiles are shown instead of reflectance profiles because they have the same shapes as the reflectance profiles but lower (i.e. near-zero) baselines to facilitate interpretation. The blank thermogram shows a constant high laser transmittance throughout the course of analysis, indicating the absence of light absorbing materials on the blank filter. The thermogram of the sample with correction shows a lower starting laser transmittance, indicative of the amount

of light absorbing materials on the filter, and exhibits a U-shaped trend as OP was formed and accumulated in the inert stage and later liberated in the oxidizing stage; the split between OC and EC was determined as the point when the laser transmittance rose back to its initial level, indicating complete oxidation of OP. Toward the end of the analysis, the recovery of laser transmittance to a level comparable to that of the blank filter indicated fully evolved carbon from

the filter. By comparison, the thermogram of the sample without correction exhibits a number of different attributes. First, the initial transmittance is much lower at a near-zero level, meaning that the laser illumination is completely attenuated due to the presence of strong absorbing materials on the filter; filters with optical profiles like this are black in color (not shown). As analysis time elapsed and the program advanced to higher temperature set-points, the laser signal

remained almost unchanged until it started to rise slightly at high oxidizing temperatures (740-





840°C). The final laser transmittance level was much lower than those of the normal or the blank sample, indicating substantial unevolved EC remaining on the filter. For the sample without correction, the OC-EC split point was determined as the point when the system switched to the oxidizing stage, the same as the split for $OC_{sum}$ and $EC_{sum}$. In these cases, the complete

attenuation of laser signal led to insufficient dynamic range for it to respond to carbon pyrolysis, regardless of how much OP was formed.

The initial and final readings of laser transmittance are compared among the three groups of samples, i.e., "blank" (n = 512), "OP > 0" (n = 3894), and "OP = 0" (n = 179), in Figure 6b-6e. Despite the variations within each of the three groups due to uncontrollable factors (e.g.,

different units of the same TOA model), those aforementioned desirable attributes of the analysis thermograms of the "OP > 0" and "blank" groups are statistically evident, including the consistency between initial and final laser transmittance for the blank samples, as well as the closeness of the final laser transmittance to the blank levels for the "OP > 0" group. Also evident were the distinctly different patterns of both initial and final laser transmittance distributions of

the "OP = 0" group compared to the "OP > 0" group. Low initial and final transmittance readings were observed for the "OP = 0" group, with the former close to the laser detector baseline and the latter remaining well below the blank levels.

These results led to the following conclusions. First, for ~5% of the CSN quartz filter samples, undetected OP, and hence no optical charring correction, likely resulted from complete

attenuation of the laser signal, leading to large inter-model discrepancies in EC between Sunset and DRI-2015. Second, EC mass loadings from these samples were likely underestimated by both models, as suggested by residual EC unevolved from the filters at 840°C, the highest IMPROVE_A temperature plateau.

### 3.2.2 Instrument differences causing carbon migration

The results presented in Sect. 3.1.2 show notable inter-model differences in the OC and EC subfractions, or carbon migration, caused by differences in instrument configurations between Sunset and DRI analyzers. Diagnosis and comparisons of these instrumental differences are beyond the scope of this work. In the following, we qualitatively discuss the roles of some possible factors to help formulate targeted experimental studies aimed at probing and reconciling

such differences.

Chow et al. (2015) reported similar inconsistencies when comparing the subfractions between the two DRI models and attributed such discrepancy to the variability (up to a factor of two) in the trace oxygen levels in the oven of the DRI analyzers (Chow et al., 2007), as well as slight differences in the sample temperatures. In our study, when DRI and Sunset analyzers were

compared, any difference in the sample temperatures likely resulted not only from the accuracy of the temperature calibration devices, which was typically ±1-2% of the specified temperatures (Chow et al., 2005, Phuah et al., 2009), but also from the different temperature calibration methods used by these models. As detailed in Section 2.1, Sunset analyzers use an external thermocouple that measures filter temperature and DRI analyzers use color-changing chemicals

(i.e. Tempilaq° G) to adjust the oven temperature readings at the IMPROVE_A temperature setpoints. Although a previous study by Phuah et al. (2009) demonstrated good comparability between the two temperature calibrations, the external calibration thermocouple in the Sunset analyzer used in that study was modified from those in the commercially available temperature calibration kit (Sunset Laboratory, Inc, OR, US) used in the present study. Chow et al. (2005)

found that lowering sample temperatures by 14 to 22°C in the IMPROVE protocol reduced OC1-OC3 subfractions and increased OC4, OP and EC subfractions. In our results, the inter-model differences in OC1, OC3 and OC4 were in the same direction, opposite to the differences in OC2 and EC subfractions, suggesting that either the temperature differences between models at each set-point were not in the same direction or temperature differences alone cannot fully explain the

observed subfraction migration.

Choices made in determining thresholds for peak integration (baselines) and times spent at each temperature set-point are also expected to contribute to the observed inter-model differences in the subfractions. UCD-Sunset data processing integrates all carbon signals above the detection baseline, which is calculated as the average FID count during the first 10 seconds of an analysis.

In contrast, DRI models set a peak integration threshold, specific to the detector type, on top of the detection baseline to account for baseline drift and noise. DRI-2001 used a flat FID count of 1 on top of the baseline as the peak integration threshold, which translates to a threshold of 0.72 (ppm $CO_2 \times$ mL/min) used in DRI-2015 with an NDIR detector (a threshold of 2 is used for EC3 peak integration) (Chow et al., 2017). The difference in the integration threshold will have the

most impact on the subfractions with lower mass loadings. In addition, since the analysis time in the IMPROVE_A protocol is concentration-driven, the program advances to the next temperature step either when the carbon signal from the current step returns to the baseline level or when the duration of the current step exceeds 580 seconds. A higher baseline would result in earlier advance of the temperature step, leading to less carbon evolved during that step. This

effect is likely to have a more significant impact on samples with higher ambient concentrations (Zheng et al., 2014).

## 4. Conclusions and implications

A detailed study is performed to assess the inter-model differences among the three models of

carbon analyzers used for CSN TOA carbon analysis during the past decade (2010-2019). Two sets of CSN quartz filter samples were used for comparison, each analyzed by a pair of the three analyzer models. The first set includes 4073 samples and 622 field blanks collected in 2017, sequentially analyzed by the Sunset and DRI-2015 analyzers within a year. The second set consists of 303 archived samples collected in 2007, originally analyzed by the DRI-2001

analyzers in 2008 and reanalyzed by the Sunset analyzers in 2017-2018. By using the same IMPROVE_A protocol, these two comparisons allow for a focused examination of instrumentation differences, especially those between the Sunset and DRI analyzers.

Our results provide quantitative evidence of desirable consistency in TC and the major carbon fractions (OC and EC), with mean scaled relative differences (SRDs) within 2% for TC, 5% for

OC, and 12% for EC, along with high correlation coefficients above 0.95 for TC and OC, and above 0.90 for EC. Underlying the consistency in bulk carbon fractions were relatively larger and diverse inter-model differences in OC1-OC4, EC1-EC3 and OP subfractions. Better inter-model agreement was found for subfractions with relatively high mass loading and smaller within-model uncertainties (e.g., OC2, OC3, and EC1). Sunset EC subfractions were consistently

higher, with SRDs varying from 5.4% for EC1 between Sunset and DRI-2001 up to 137% for EC3 between Sunset and DRI-2015. Pyrolyzed carbon (OP) formation from charring is found to be highly instrument dependent, differing by 38% and 66% in mean SRD between Sunset and DRI-2001 and between Sunset and DRI-2015, respectively. The observed migration among the



thermal subfractions is likely related to slight differences in the instrument thermal parameters and configurations, such as sample temperature, baseline selection and residence time, between Sunset and DRI analyzers.

Optical charring correction reduced the inter-model biases in OC and EC relative to those of $OC_{sum}$ (OC1+OC2+OC3+OC4) and $EC_{sum}$ (EC1+EC2+EC3) by 56%- 67% and 75%-76%, respectively. The remaining inter-model discrepancy in EC was found to be substantially larger

for ~5% of the 2017 CSN samples that had no instrumentally detected OP. Examination of Sunset analysis thermograms suggested that complete laser signal attenuation was the cause; such samples occur more frequently at higher EC mass loadings and were often associated with residual EC that was resistant to the highest IMPROVE_A temperature plateau (840°C), suggesting that both models might underestimate the true ambient EC concentrations for a subset

of CSN samples. Previous study by Han et al. (2007) found that EC originated from diesel sources had higher decomposing temperature than EC from biomass burning. Since the vast majority of CSN sites are located in urban areas (Solomon et al., 2014), where the sampled air is heavily impacted by anthropogenic emissions, it is possible that the samples with no instrumentally detected OP were heavily influenced by diesel fuel combustion.

Our work offers comprehensive information on TOA instrument uncertainty and inter-model differences necessary for future studies to consider in assessing long-term trends in CSN carbon data. Such information will also assist performance evaluation of chemical transport models using CSN data. Additionally, inter-model differences in thermal subfractions of OC and EC shown here suggest source apportionment studies on multi-year trends that utilize TOA thermal

subfractions as input data in source profiles (e.g. Kim and Hopke, 2005) need to take into consideration the consistency and comparability of data from different carbon analyzer models.

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



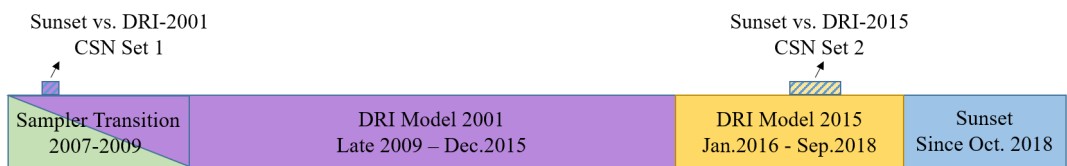

**Figure 1.** Timeline of CSN network-wide changes in carbon analysis from 2007 to present, including changes in sample collection and analytical protocol during the "Sampler Transition" period from 2007 – 2009, as well as two instrumentation changes in 2016 and late 2018. The approximate sample date range of the CSN 2007 (CSN Set 1) and 2017 (CSN Set 2) filter sets used in the inter-model comparisons are marked on the timeline.






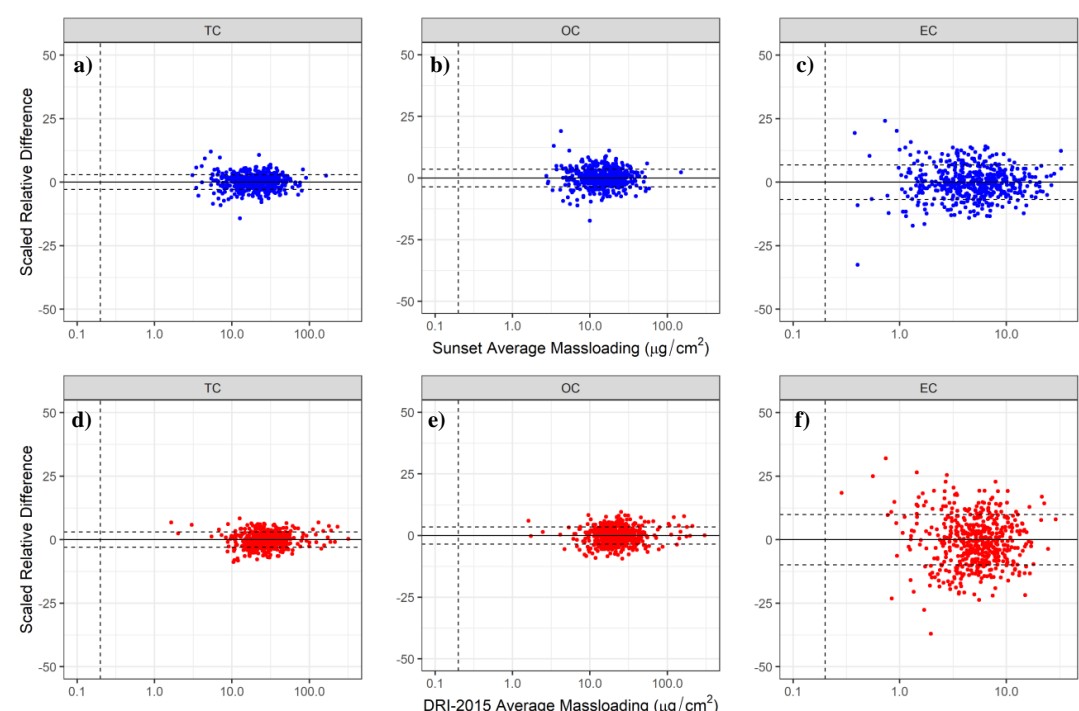

**Figure 2**. Scaled relative difference (equation 1, %) of TC, OC and EC, calculated from the original and replicate paired analyses performed on 2017 CSN samples by the Sunset analyzers (a-c) and the DRI Model 2015 analyzers (d-f), as a function of the average mass loading between the two analyses. The horizontal dashed lines in each plot represents ±1σ of the SRD determined for each carbon component. The vertical dashed line intercepted at mass loading of 0.2 µg/cm² indicates the method detection limit (MDL).






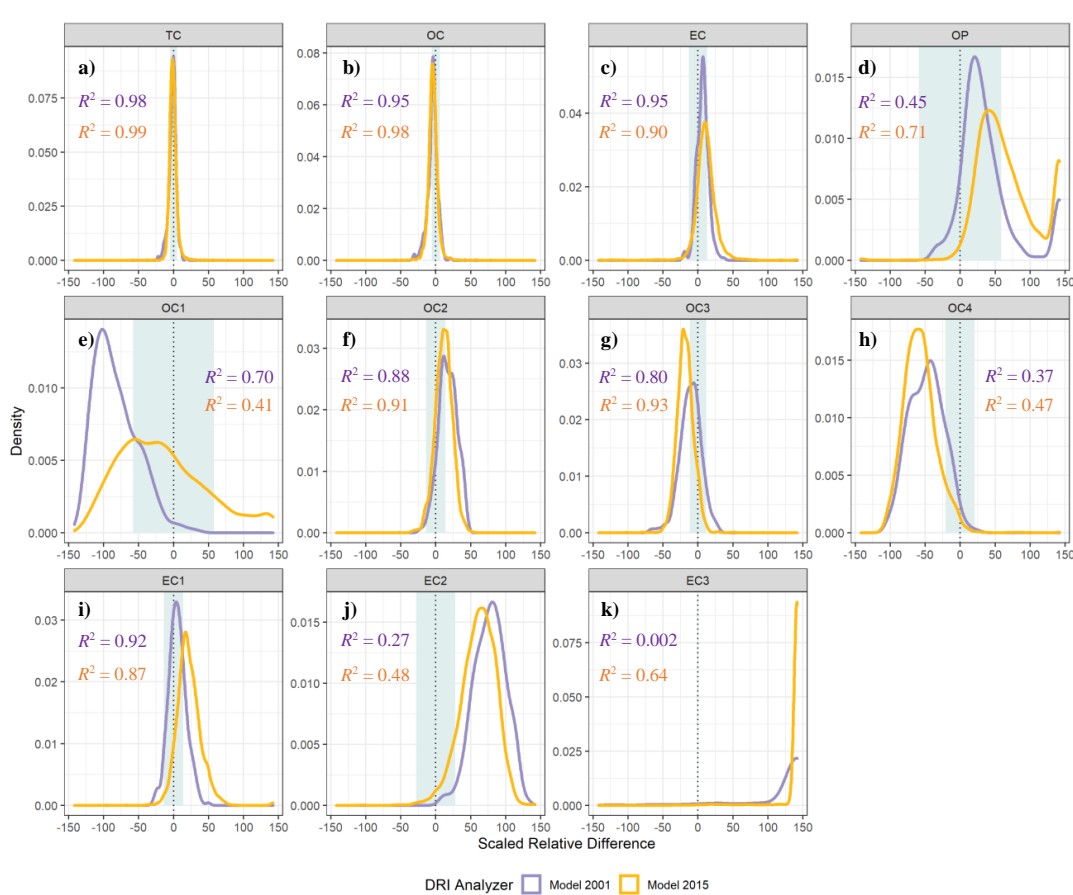

**Figure 3.** Probability density curves of scaled relative differences (equation 4, %) between the Sunset analyzer versus two DRI analyzers for all carbon components (a-k). Yellow lines (and text) denote the CSN 2017 samples analyzed by DRI Model 2015 and Sunset, whereas the purple lines (and text) denote the archived CSN 2007 samples analyzed by DRI Model 2001 and Sunset. $R^2$ values are derived from linear regression of each dataset. The gray shaded area indicates the inter-model uncertainty (equation 2a, %) for each carbon component except for EC3 (Table 3).




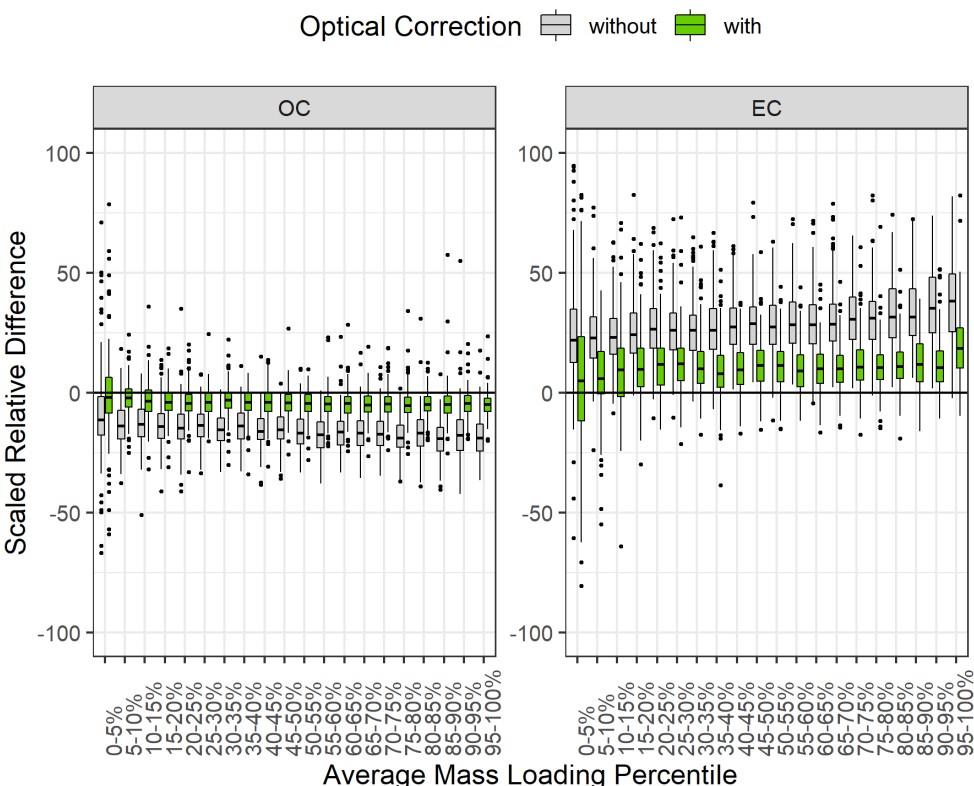


**Figure 4**. Distribution of scaled relative difference between Sunset and DRI Model 2015 in OC and EC without optical correction (i.e., $OC_{sum}$ and $EC_{sum}$, grey boxes) and with optical correction (green boxes) for each 5th percentile bin of its average mass loading. The thick horizontal lines indicate median, and the upper and lower limits of the boxes represent 75th and 25th percentile, 665 respectively. The whiskers extend to 1.5×IQR (where IQR is the interquartile range, or the distance between the 25th and the 75th percentiles). Outliers are shown as black dots.





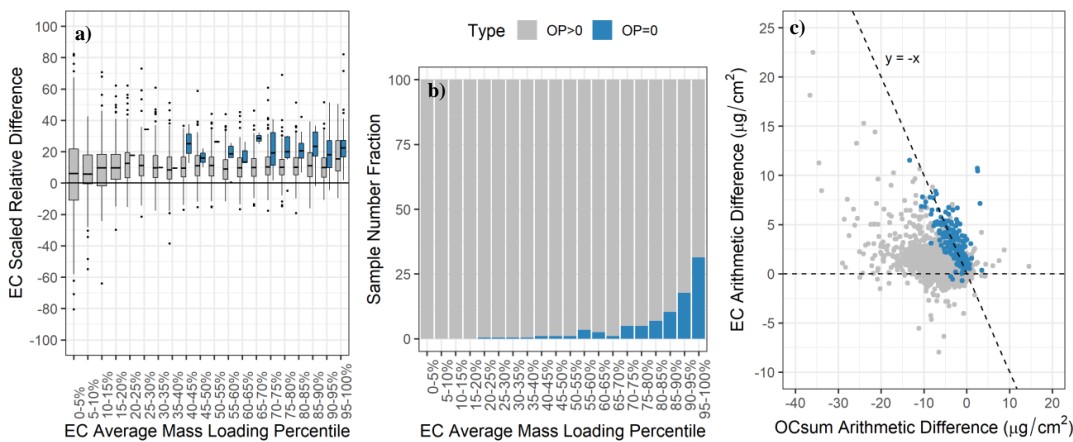


**Figure 5.** a) Distribution of scaled relative difference in EC between Sunset and DRI Model
2015 for samples with (i.e. OP > 0) and without (i.e. OP = 0) optical charring correction on both
analyzers, b) fraction of samples that had no charring correction (i.e. OP = 0) in each EC mass
loading bin, and c) scatter plot of arithmetic difference in EC vs arithmetic difference in OC$_{sum}$

between Sunset and DRI Model 2015.



**Figure 6.** (a) Example laser transmittance response at different thermal-optical analysis stages
(i.e. inert, oxidizing and cool-down) for a blank, a CSN sample with optical charring correction
(i.e. OP > 0) and a CSN sample with no optical charring correction (i.e. OP = 0); (b) and (d)
Cumulative plot and histogram of the laser transmittance initial readings, and (c) and (e)
cumulative plot and histogram of the laser transmittance final readings for all blanks ($n = 512$)
and CSN samples with ($n = 3894$) and with no optical charring correction ($n = 179$). All laser
readings are in arbitrary unit (a.u.) and are from Sunset analyzers.




**Table 1.** Key attributes of three analyzer models (DRI Model 2001, DRI Model 2015 and Sunset analyzers) in comparison.

|  | DRI Model 2001 | DRI Model 2015 | Sunset Model 5L |
|---|---|---|---|
| Laser Source | Helium-neon (He-Ne) laser at 633 nm | Seven diode lasers at 405, 445, 532, 635, 780, 808, and 980 nm | Single diode laser at 658 nm |
| Detection | Flame ionization detector (FID) for $CH_4$ | Non-dispersive infrared (NDIR) detector for $CO_2$ | Flame ionization detector (FID) for $CH_4$ |
| Temperature Calibration | Temperature-indicating liquids (Tempilaq° G) that change optical properties at 121°C, 184°C, 253°C, 510°C, 704°C and 816°C to calibrate oven temperature | Same as DRI Model 2001 | A thermocouple at sample position to calibrate oven temperature at 140°C, 280°C, 480°C, 580°C, 740°C and 840°C (IMPROVE_A temperature plateaus) |
| Optical Configuration | Laser source installed coaxially with the optical detectors; laser beam travels in optical fiber and then through quartz guiding pipe before reaching the sample. | Same as DRI Model 2001 | Laser source installed diagonally to the optical detectors with a 45° angle; laser beam travels through quartz oven window in carrier gas before reaching the sample. |





**Table 2.** Mean and standard deviation (1σ) of the carbon mass loadings (in µg/cm$^2$) from 622
field blank measurements by the Sunset and DRI Model 2015 analyzers.

| Carbon Component | Sunset | DRI Model 2015 |
|---|---|---|
| | Mean (± one standard deviation) | |
| Total Carbon (TC) | 2.2 (± 1.2) | 1.5 (± 0.9) |
| Organic Carbon (OC) | 2.1 (± 1.1) | 1.4 (± 0.7) |
| Elemental Carbon (EC) | 0.1 (± 0.3) | 0.0 (± 0.2) |
| Pyrolyzed OC (OP) | 0.2 (± 0.3) | 0.0 (± 0.0) |
| OC1 | 0.7 (± 0.6) | 0.2 (± 0.2) |
| OC2 | 0.6 (± 0.2) | 0.4 (± 0.2) |
| OC3 | 0.5 (± 0.4) | 0.7 (± 0.4) |
| OC4 | 0.2 (± 0.2) | 0.1 (± 0.2) |
| EC1 | 0.1 (± 0.3) | 0.0 (± 0.2) |
| EC2 | 0.1 (± 0.1) | 0.0 (± 0.1) |
| EC3 | 0.0 (± 0.1) | 0.0 (± 0.0) |



**Table 3.** Within-model replication bias and uncertainty estimated from the scaled relative difference of the replicate analyses by Sunset and DRI Model 2015 analyzers, as well as the inter-model uncertainty calculated from the within-model replication uncertainties in the individual models.

| Carbon Component | Sunset (*n* = 519) | | DRI Model 2015 (*n* = 518) | | Inter-model |
|---|---|---|---|---|---|
| | Bias (%) | Uncertainty (%) | Bias (%) | Uncertainty (%) | Uncertainty (%) |
| Total Carbon (TC) | 0.0 | 2.9 | -0.2 | 3.0 | 4.1 |
| Organic Carbon (OC) | 0.1 | 3.6 | 0.0 | 3.5 | 5.0 |
| Elemental Carbon (EC) | -0.1 | 6.8 | -0.9 | 9.7 | 12 |
| Pyrolyzed OC (OP) | 1.3 | 13 | 0.8 | 56 | 58 |
| OC1 | 0.4 | 27 | -2.2 | 50 | 57 |
| OC2 | 0.1 | 7.9 | 0.6 | 10 | 13 |
| OC3 | -0.3 | 7.7 | 0.0 | 7.2 | 11 |
| OC4 | -0.4 | 16 | 0.6 | 11 | 20 |
| EC1 | 0.3 | 6.6 | -0.4 | 11 | 13 |
| EC2 | 0.0 | 16 | -2.7 | 22 | 27 |
| EC3 | NA[*] | | | | |

*Too few (less than 20%) data points have mass loadings that are greater than 3 times the MDL





698
699
700

**Table 4.** Mean and standard deviation ($1\sigma$) of the average mass loading (in $\mu g/cm^2$), arithmetic mass loading difference (in $\mu g/cm^2$) and scaled relative difference (dimensionless) between the Sunset vs. DRI Model 2001 pair (left columns) and the Sunset vs. DRI Model 2015 pair (right columns) for bulk carbon components and their thermal fractions.

| Carbon Fraction | Sunset (Y) to DRI Model 2001(X) (n = 303) | | | Sunset(Y) to DRI -2015(X) (n = 4073) | | |
|---|---|---|---|---|---|---|
| | Average Mass Loading $\frac{Y+X}{2}$ ($\mu g/cm^2$) | Arithmetic Difference $Y-X$ ($\mu g/cm^2$) | Scaled Relative Difference $\frac{(Y-X)/\sqrt{2}}{(Y+X)/2}\times100$ | Average Mass Loading $\frac{Y+X}{2}$ ($\mu g/cm^2$) | Arithmetic Difference $Y-X$ ($\mu g/cm^2$) | Scaled Relative Difference $\frac{(Y-X)/\sqrt{2}}{(Y+X)/2}\times100$ |
| | | | Mean (± one standard deviation) | | | |
| Total Carbon (TC) | 31 (± 18) | -1.0 (±2.8) | -1.6 (± 5.4) | 27 (± 19) | -0.5 (± 2.0) | -0.9 (± 6.0) |
| Organic Carbon (OC) | 22 (± 12) | -1.7 (± 3.0) | -4.6 (± 7.2) | 21 (± 16) | -1.4 (± 2.4) | -4.1 (± 7.1) |
| Elemental Carbon (EC) | 9.4 (± 6.1) | 0.7 (± 1.5) | 6.5 (± 8.3) | 5.5 (± 3.7) | 0.9 (± 1.4) | 11 (± 15) |
| Pyrolyzed OC (OP) | 4.7 (± 3.0) | 2.1 (± 2.8) | 38 (± 41) | 3.4 (± 4.3) | 2.9 (± 2.7) | 66 (± 41) |
| OC1 | 2.0 (± 2.2) | -2.3 (± 2.1) | -82 (± 31) | 1.1 (± 1.3) | -0.5 (± 1.4) | -16 (± 63) |
| OC2 | 6.8 (± 3.8) | 1.7 (± 1.7) | 17 (± 13) | 5.9 (± 4.4) | 0.8 (± 1.4) | 11 (± 12) |
| OC3 | 5.2 (± 3.1) | -0.7 (±1.5) | -11 (± 15) | 7.7 (± 5.6) | -1.7 (± 1.5) | -18 (± 12) |
| OC4 | 3.3 (± 1.6) | -2.6 (± 2.3) | -48 (± 24) | 3.2 (± 2.0) | -2.8 (± 2.3) | -58 (± 24) |
| EC1 | 13 (± 8.1) | 1.3 (± 2.7) | 5.4 (± 13) | 7.5 (± 6.2) | 2.6 (± 3.0) | 22 (± 18) |
| EC2 | 1.3 (± 0.7) | 1.5 (± 1.0) | 77 (± 23) | 1.3 (± 0.7) | 1.1 (± 0.6) | 61 (± 25) |
| EC3 | 0.1 (± 0.0) | 0.1 (± 0.1) | 122 (± 51) | 0.1 (± 0.2) | 0.1 (± 0.2) | 137 (±23) |
| $OC_{sum}$ = OC1+OC2+OC3+OC4 | 17 (± 9.7) | -3.8 (±3.8) | -14 (±10.2) | 18 (± 13) | -4.3 (± 4.0) | -16 (± 9.1) |
| $EC_{sum}$ = EC1+EC2+EC3 | 14 (± 8.4) | 2.8 (±2.6) | 14 (±9.3) | 8.9 (± 6.5) | 3.8 (± 3.2) | 29 (± 15) |