# Peer review of "Inter-Comparison of Thermal-Optical Carbon Measurements by Sunset and DRI Analyzers Using the IMPROVE\_A Protocol"

_Atmospheric Measurement Techniques, 2020_

## Referee Comment (RC1) · Anonymous Referee #1 · 9 Dec 2020

Review of" Inter-Comparison of Thermal-Optical Carbon Measurements by Sunset and DRI Analyzers Using the IMPROVE_A Protocol" by Zhang et al.

The manuscript (amt-2020-436) reported an inter-instrument comparison between two pairs of carbon analyzers using two large datasets from CSN. The reviewer fully agreed that these comparisons are valuable since inter-instrument comparisons remain limited in the literature. The authors present substantial analyses, but some analyses seem incomplete. The reviewer feels that the data analysis and interpretation can be improved, otherwise, the value of this work could be weakened. The following issues should be addressed before consideration of publication.

1) Line 65-69. Please elaborate on why the carbon analyzer was changed from DRI2015 to Sunset 5L in 2018? The transition from DRI2001 to DRI2015 in 2016 is easy to understand, but the 2018 instrument change was too close to the 2016 instrument change. The readers would wonder about the motivation behind the 2018 instrument transition.

2) Section 2.1 was well written. This section provides a useful description of the three instruments, which covered all key features of instruments from the engineering perspective. That is very helpful and vital for the readers to understand the differences in instrument design.

3) Line 180. The author stated that the mass loading of DIR2001 data was back-calculated from the air concentration data using a constant sample volume (33 m$^3$). That would introduce unnecessary uncertainties for the instrument comparison since the actual sample volume varied by samples. The deviation of sample volume could be small, but using a constant sample volume for back-calculation is not scientifically sound. The author should use the actual sample volume for the back-calculation of mass loading of DIR2001 data.

4) The reviewer suggests comparing the average temperature profile between DRI and Sunset instruments for the two sets of data. First, there could be differences in the actual temperature for a specific temperature plateau when implementing the same TOA protocol. In addition, since temperature ramping rates were not defined in most TOA protocols, the differences in temperature ramping rates between the three carbon analyzers could be one source of carbon fraction discrepancy and are worth further investigation.

5) Line 380-381. The low laser transmittance (T) signal after analysis indicates the remaining strong light-absorbing materials on the filter. These materials are refractory, but not necessarily to be EC. One possibility is metal oxides as shown in the figure below. The authors are encouraged to check the elements data to see if there are any correlations between crustal elements abundance (e.g. Fe) and the abnormal low laser T signal after analysis. If yes, that would be an indication of metal oxide's influence.

[Figure]

**Figure R1.** A photo of filters after OC/EC analysis. The red circle highlights an example of abundant metal oxides on the quartz filter after OC/EC analysis, leading to browning of the filter.

6) Section 3.2.1. The reviewer believed the OP mentioned here is reflectance-based. However, Figure 5&6 rely heavily on laser transmittance readings. It is suggested to use OP_R instead of OP to avoid any confusion that may cause.

7) Section 3.2.1 needs more in-depth analysis. The analysis presented here shows that low initial laser T (by Sunset analyzer) is associated with OP_R=0. There could be two possibilities leading to low initial laser T. One possibility (case 1) is a result of high native EC on the filter. That was likely a result of the high surface loading by the IMPROVE sampler, which was mainly designed for remote locations with low PM concentrations. The IMPROVE sampler (22.8 L/min) has a higher flow rate comparing to the SASS sampler (6.7 L/min), which was previously used by CSN before the sampler transition. Besides, the deposit filter area of the IMPROVE sampler (3.53 cm$^2$) was smaller than the SASS sampler (11.3 cm$^2$). As a result, the surface loading of the IMPROVE sampler was higher than SASS sampler by a factor of 11, leading to a higher occurrence of low initial laser T. The situation was worsened for CSN samples, which were coming from urban sites with high PM loadings. That poses a challenge for the optical correction in TOA. These should be mentioned and discussed. The second possibility (case 2) is due to high light-absorbing metal oxides on the filter. It is worth investigating the contribution of different causes. For example, how many percentages from case 1, case 2, and both cases 1&2? It would be interesting to see how often that case 1 and case 2 occurred at the same time.

8) Following the last question, as shown in Figure 6d, for the low initial laser T samples (say initial laser T by Sunset analyzer<1000), it seems that half of the samples end up with OP_R=0, but the other half end up with OP_R>0. Why these low initial laser T samples end up with different OP_R? Considering that initial laser T was already low, further darkening of the filter due to charring was likely out of the dynamic range of the optical system, why some samples still get OP_R>0?

9) The authors may consider providing a recommended threshold on the useable range of the laser reading (could be on an absolute or a relative scale). That would be a useful indication for the possible saturation of the optical system due to high loading samples.

10) Section 3.2.2. The authors mentioned that the duration of each step in the IMPROVE_A protocol was concentration-driven. It would be interesting to examine the distribution of duration difference of each carbon fraction between Sunset and DRI instruments. That could be one of the possible sources of carbon fraction divergence between the two instruments.

Technical comments:

1) Line 41. Please cite the latest version of the IPCC report (AR5).
2) Figure 2. Please specify the sample numbers of each plot. Please also add DRI2001 and DRI2015 annotations directly on each plot for easy reference.
3) Figure 6 d&e. Try normalized histograms. The sample size is quite different for the three groups. It is difficult to see the distribution, especially for Figure 6e.

---

## Referee Comment (RC2) · Anonymous Referee #2 · 9 Dec 2020

Review of "Inter-Comparison of Thermal-Optical Carbon Measurements by Sunset and DRI Analyzers Using the IMPROVE_A Protocol" by Zhang et al.

Review:

This paper discussed the OCEC measurements inter-comparison results obtained from the Chemical Speciation Network (CSN).  There are two sets of inter-comparison, but in general, they are comparison between OCEC results obtained from a Sunset instrument vs. DRI analyzer. The main difference between the two sets of measurements lies in the technical details, which are the different analyzer model and the date coverage of the measurements.  This is supposed to be a useful inter-comparison exercise and will help the network to understand the impact when switching instrument and how the long-term measurements can be interpret when using it for trend analysis.  This work certainly is valuable, however, I find it needs a major revision to re-organize the paper better before it can be published.  This is rather confusing and I don't think there is enough interpretation regarding the actual comparison between the two sets of measurements in a reader perspective.  There are many aspects the authors are presenting in this paper, but they are not presented in a way that is easy to follow.  I have summarized a list of things that can help improve this manuscript before it is published in AMT.

General comment:

In this paper, the authors rely a lot of statistics and try to interpret the difference (or bias) between the two data sets looking at all data as a population.  It is one way of looking at things and to get a sense of the magnitude of this bias in all measurements as a whole.  But for ambient measurements, sometimes it is also important to understand the time series of this deviation and to really understand the reason behind the bias.  For example, does the bias is higher in summer than winter?  Or is always consistent over the course of the sampling period?  During the analysis, the authors discovered a significant number of samples that have no observable or detectable "OP". More investigation is needed to understand if this may impact the data differently for example over different seasons.

In order to obtain the correct OC and EC, charred OC has to be taken into account.  The Sunset instrument laser detector detects the transmittance whereas the DRI analyzer has the option to use either transmittance or reflectance.  I could not find where the authors discuss about whether they are using transmittance or reflectance to determine charred OC (or OP).

Also, in multiple places when OC and EC were discussed, are they both corrected for charred OC?  Any comparison for OC and EC between the two data sets without accounting for charred OC is meaningless.

The authors introduce a new term, referred to as "SRD" or scaled relative difference.  I understand this may have some statistical value, but this is not a typical common term that all readers will connect to when they read.  This is particularly an issue when readers looking at various graphs generated from this value and try to understand or interpret its meaning.  One way to improve this (if the authors insist of keeping this analysis) is to do a better job in explaining this term other than by defining it again.  In other words, how readers should interpret this term for its magnitude.  For example, when one express absolute difference between two numbers, readers understand the value of 0 means there is no bias. Or

large positive or negative values means certain set is higher or lower. In comparison, SRD (because of the way it is defined), it does not easily translate.

Also, is this SRD term mostly used for interpreting the comparison between measurements with replicates? It is not clear from the heading of the sub sections to me.

Specific comment:

p.7, line 275-285. It is observed that Sunset TC is generally agree with DRI TC, however, Sunset OC was found lower than DRI OC (and the opposite is true for EC). This is obviously due to the difference in OCEC split but is could also due to the fact that one instrument used transmittance to determine charred OC whereas the other one use reflectance. I couldn't find where this is discussed or whether the authors are using data to ensure the consistency (i.e., all corrected data are based on transmittance or reflectance).

p.9, line 339. Determining the correct OC and EC really requires the correction of OP. So what is the purpose of showing the "uncorrected" OC and EC when you know OP has to be taken into consideration?

Also, by definition, OP is subtracted from EC and added to OC to determine the final EC and OC. Therefore, in comparing the "uncorrected" and "corrected" OC, one should expect OC and EC be shifted in the same magnitude but in opposite direction. How come this does not seem to be the case in Fig. 4? Is this because it is plotted as a "relative" difference rather than absolute difference? Would absolute difference be more meaningful than the relative difference in this case?

The data in Fig. 4 are sorted according to the "mass loading percentage". How is this defined? Do the authors mean the amount of OC or EC relative to TC? So what is the advantage of plotting the data this way rather than sorting them by the absolute EC and OC mass?

p.9, line 348-350. The authors suggested for EC measurements with large SRD were samples with no instrumentally detected OP. What is considered "large"? Better to give a range or value or threshold. Does this represent bars with high "relative difference" in Fig 4?

In addition, the authors said there are consider number of samples with no detectable OP, if I understand this correctly, no "OP" means the "corrected" OC (or EC) and "uncorrected" OC (or EC) are equal, correct? However, I don't ever see any evidence that there are any samples with "non-detectable OP" in Fig. 4. Fig. 4 shows that there are always a considerable amount of OP in all samples!

p.9, line 355-359. The authors suggested that the EC bias almost solely originated from the thermal effect and I don't understand the argument supporting this statement. In Fig. 5, I only see the sub group of measurements with "no OP" (i.e., no thermal correction) has even higher relative difference than the group with considerable amount of OP. Or may be I don't understand how to interpret this graph. Plotting in relative difference could be one issue. If the authors' statement is true, I would expect for the sub-group of measurements with no OP should give you no bias between the two data set. Is this the case?

p.9, line 360-386.  This whole paragraph may be should not belong in main text.  Probably is better to be included in supplementary information and then just refer to it in the main text.  This paragraph is mostly to explain Fig. 6a, however, I would suggest the authors also include the detector signal, so the authors would better understand how the OP peak compared to OC or EC and aid the interpretation of the thermograms.  Only the laser response in Fig. 6 does not help much.

Fig. 6b to e.  What is the purpose of these?  How does the different shapes in Fig. b and c are supposed to mean and what we should expect?  I really don't think these graphs really aid much in terms of understanding what is going on regarding the "OP = 0" issue.  The authors can however keep them in supplementary info if they want.  What would really help is to actually include a completed thermograms including both laser response and detector response.  Only that will allow the readers to understand how the samples evolve over the course of the analysis in a typical situation when OP=0 and OP>0 and may also include the case for blank as reference.

---

## Author Comment (AC1) · 9 Mar 2021

Reviewer 1's comments

The manuscript (amt-2020-436) reported an inter-instrument comparison between two pairs of carbon analyzers using two large datasets from CSN. The reviewer fully agreed that these comparisons are valuable since inter-instrument comparisons remain limited in the literature. The authors present substantial analyses, but some analyses seem incomplete. The reviewer feels that the data analysis and interpretation can be improved, otherwise, the value of this work could be weakened. The following issues should be addressed before consideration of publication.

We thank the reviewer for his/her comments on our paper. In the following we provide our point-to-point responses, as well as list our revisions to the text to address these comments.

1) Line 65-69. Please elaborate on why the carbon analyzer was changed from DRI2015 to Sunset 5L in 2018? The transition from DRI2001 to DRI2015 in 2016 is easy to understand, but the 2018 instrument change was too close to the 2016 instrument change. The readers would wonder about the motivation behind the 2018 instrument transition.

The change in the carbon analyzer in 2018 is a result of a change in the analytical laboratory that does CSN carbon analysis. In the revised manuscript (Line 63-69) we have added the following text to briefly explain the instrument transition:

*"As shown in Figure 1, in the beginning of 2016, TOA carbon analysis for CSN transitioned from using the Desert Research Institute (DRI) Model 2001 analyzers (termed "DRI-2001" hereinafter) to DRI Model 2015 multi-wavelength analyzers (termed "DRI-2015" hereinafter) as a result of instrument upgrade, and again in October 2018, CSN TOA transitioned from using DRI-2015 analyzers to Sunset Laboratory Model 5L analyzers (termed "Sunset" hereinafter) due to change in the analytical laboratory (from DRI to UC Davis)."*

2) Section 2.1 was well written. This section provides a useful description of the three instruments, which covered all key features of instruments from the engineering perspective. That is very helpful and vital for the readers to understand the differences in instrument design.

We thank the reviewer for the positive feedback.

3) Line 180. The author stated that the mass loading of DRI2001 data was back-calculated from the air concentration data using a constant sample volume (33 m$^3$). That would introduce unnecessary uncertainties for the instrument comparison since the actual sample volume varied by samples. The deviation of sample volume could be small, but using a constant sample volume for back-calculation is not scientifically sound. The author should use the actual sample volume for the back-calculation of mass loading of DRI2001 data.

We agree that sample-specific volume data, if exists, would be superior to a constant volume to use in calculating mass loadings from concentrations. Nevertheless, such data was unavailable and therefore a constant sample volume was used instead. The uncertainty as a result of this is small given the stringency of CSN operational tolerances for flow rate and sample duration. In CSN, sample is invalidated if the sample flow rate is outside 10% of the nominal flow rate and/or if the sample time is more than 25 hours or less than 23 hours.

We have added a statement in the revised text (Line 197-199) to acknowledge this uncertainty: "*The use of nominal instead of the actual sample volume adds little uncertainty, given the stringency of CSN operational tolerances for flow rate and sample duration.*"

4) The reviewer suggests comparing the average temperature profile between DRI and Sunset instruments for the two sets of data. First, there could be differences in the actual temperature for a specific temperature plateau when implementing the same TOA protocol. In addition, since temperature ramping rates were not defined in most TOA protocols, the differences in temperature ramping rates between the three carbon analyzers could be one source of carbon fraction discrepancy and are worth further investigation.

We agree that such a comparison would be useful as it is true that the IMPROVE_A protocol allows for some play in details such as temperature ramping rates and criteria for advancing to the next stage. Unlike most TOA protocols, IMPROVE_A does not pre-specify the temperature profile as the same fixed function of time for each individual sample, but instead leaves it contingent on individual sample compositions and loadings. As Chow et al. (2007) explain, "Temperature is ramped to the next step when the FID [or NDIR] response returns to baseline or remains constant for more than 30 sec; the residence time at each plateau is longer for more heavily loaded samples." Unremarked differences in implicit tolerances for ramping rates, and for determining "return to baseline or remains constant", undoubtedly do contribute some of the differences we observe in different models' reported results. In this work, the time profiles of temperature and evolved carbon for individual samples are available with the Sunset instruments at UCD, but not available from the DRI analyzer. We do agree that a targeted study of such difference between Sunset and DRI analyzers in the future will further refine our understanding of its role in the differences in the analysis results.

We have revised the second paragraph in Section 3.2.2 (Line 449-458) to read "*In addition, details in instrument configuration and operating parameters set by the analysis control program, often invisible and unalterable to end users, can be distinct among TOA models from different manufacturers. As Chow et al. (2007) explain, "Temperature is ramped to the next step when the FID [or NDIR] response returns to baseline or remains constant for more than 30 sec; the residence time at each plateau is longer for more heavily loaded samples." Unremarked differences in implicit tolerances for temperature ramping rates, and for determining "return to baseline or remains constant", undoubtedly contribute some of the differences we observe in different models' reported results. Unfortunately, the time profiles of temperature and evolved carbon for individual samples are not routinely reported by DRI and were not available to us for systematic comparison with those from the Sunset instruments at UCD.*"

We have also added a statement in the revised text (Line 483-486) as an implication of this study that reads:

"*It should also be noted that the IMPROVE_A protocol allows for some play in details such as temperature ramping rates and criteria for advancing to the next stage. A targeted study of such difference between Sunset and DRI analyzers in the future will further refine the understanding of its role in the differences in the analysis results.*"

5) Line 380-381. The low laser transmittance (T) signal after analysis indicates the remaining strong light-absorbing materials on the filter. These materials are refractory, but not necessarily to be EC. One possibility is metal oxides as shown in the figure below. The authors are encouraged to check the elements data to see if there are any correlations between crustal elements abundance (e.g., Fe) and the abnormal low laser T signal after analysis. If yes, that would be an indication of metal oxide's influence.

Thanks for the comment. We agree and are aware that the presence of metal oxide on filter samples is one of the possible causes of low final transmittance/reflectance. During our routine operation of carbon analysis, samples that show orange/red color after TOR analysis due to high Fe abundance are flagged

'ME-1' (or 'm2' in DRI flagging system). Similarly, samples that show gray/black color after analysis due to residual EC are flagged 'ME-2' (or 'm5' in DRI flagging system). We have confirmed that the vast majority of the low final T/R samples have the "ME-2" flag.

We have added a statement in the revised text (Line 397-398) that reads:
*"Filters with this type of optical profile are black in color before analysis and remain gray/black after analysis."*

6) Section 3.2.1. The reviewer believed the OP mentioned here is reflectance-based. However, Figure 5&6 rely heavily on laser transmittance readings. It is suggested to use OP_R instead of OP to avoid any confusion that may cause.

We thank the reviewer for the suggestion. In order to avoid any confusions, we have revised Figure 6 (and the new Figure 7) to show reflectance response instead of transmittance. With this change, we keep OP as is, and have made it clear that the optical correction method is by reflectance by replacing "optical charring correction" with "reflectance charring correction" in places where confusion might occur.

7) Section 3.2.1 needs more in-depth analysis. The analysis presented here shows that low initial laser T (by Sunset analyzer) is associated with OP_R=0. There could be two possibilities leading to low initial laser T. One possibility (case 1) is a result of high native EC on the filter. That was likely a result of the high surface loading by the IMPROVE sampler, which was mainly designed for remote locations with low PM concentrations. The IMPROVE sampler (22.8 L/min) has a higher flow rate comparing to the SASS sampler (6.7 L/min), which was previously used by CSN before the sampler transition. Besides, the deposit filter area of the IMPROVE sampler (3.53 cm2) was smaller than the SASS sampler (11.3 cm$^2$). As a result, the surface loading of the IMPROVE sampler was higher than SASS sampler by a factor of 11, leading to a higher occurrence of low initial laser T. The situation was worsened for CSN samples, which were coming from urban sites with high PM loadings. That poses a challenge for the optical correction in TOA. These should be mentioned and discussed. The second possibility (case 2) is due to high light-absorbing metal oxides on the filter. It is worth investigating the contribution of different causes. For example, how many percentages from case 1, case 2, and both cases 1&2? It would be interesting to see how often that case 1 and case 2 occurred at the same time.

Thanks for the comment. We have added a statement to the revised text (Line 418-421) to mention the first possibility raised by the reviewer that reads: *"The high occurrence of samples with OP=0 in CSN likely results from high sampled air volume, small filter surface area, and the closeness of sampling sites to emission sources, leading to concentrated strong absorbing materials (i.e., EC) on filter samples and posing a challenge for TOA analysis.".* The second possibility can be ruled out, as discussed in our response to comment 5 above.

8) Following the last question, as shown in Figure 6d, for the low initial laser T samples (say initial laser T by Sunset analyzer<1000), it seems that half of the samples end up with OP_R=0, but the other half end up with OP_R>0. Why these low initial laser T samples end up with different OP_R? Considering that initial laser T was already low, further darkening of the filter due to charring was likely out of the dynamic range of the optical system, why some samples still get OP_R>0?

This figure is showing data from five Sunset instruments with varying laser sensitivity and intensity (also see response to the next comment). It is possible for some instrument with higher laser sensitivity and a lower baseline to detect some OP even with low initial laser T signal.

9) The authors may consider providing a recommended threshold on the useable range of the laser reading (could be on an absolute or a relative scale). That would be a useful indication for the possible saturation of the optical system due to high loading samples.

We agree that it would be useful to develop a lower laser response threshold. However, the laser response is not standardized across the analyzers, as the actual laser reading (in arbitrary unit) is dependent on not only the optical properties of the sample but also the laser intensity, the geometry of the laser/oven/detector setup, and the response of the photodiode, making it difficult to cross-compare laser response among analyzers of the same model or among different models. As an alternative, in the EPA Air Quality System (AQS) database, the CSN samples with saturated laser problems (samples with OP = 0) are attached a qualifier code ("LJ") to warn the end users that the reported OC and EC values have larger uncertainties due to laser saturation.

10) Section 3.2.2. The authors mentioned that the duration of each step in the IMPROVE_A protocol was concentration-driven. It would be interesting to examine the distribution of duration difference of each carbon fraction between Sunset and DRI instruments. That could be one of the possible sources of carbon fraction divergence between the two instruments.

Please see our response to comment 4.

Technical comments:
1)  Line 41. Please cite the latest version of the IPCC report (AR5).

    Citation has been updated.

2)  Figure 2. Please specify the sample numbers of each plot. Please also add DRI2001 and DRI2015 annotations directly on each plot for easy reference.

    Sample numbers have been added to the captions of both Figure 2 and 3 (please note that Figure 2 doesn't include data from DRI-2001). The legend of Figure 3 (which is the DRI model annotation) has been moved to the top of the figure for easier referencing.

3)  Figure 6 d&e. Try normalized histograms. The sample size is quite different for the three groups. It is difficult to see the distribution, especially for Figure 6e.

    Figure 6b and 6c are essentially the normalized distribution. In view of the comment, we have removed Figure 6d and Figure 6e which do not add additional information to Figure 6b and Figure 6c (new Figure 7a and 7b).

---

## Author Comment (AC2) · 9 Mar 2021

Reviewer 2's comments:

Review:
This paper discussed the OCEC measurements inter-comparison results obtained from the Chemical Speciation Network (CSN). There are two sets of inter-comparison, but in general, they are comparison between OCEC results obtained from a Sunset instrument vs. DRI analyzer. The main difference between the two sets of measurements lies in the technical details, which are the different analyzer model and the date coverage of the measurements. This is supposed to be a useful inter-comparison exercise and will help the network to understand the impact when switching instrument and how the long-term measurements can be interpreted when using it for trend analysis. This work certainly is valuable; however, I find it needs a major revision to re-organize the paper better before it can be published. This is rather confusing, and I do not think there is enough interpretation regarding the actual comparison between the two sets of measurements in a reader perspective. There are many aspects the authors are presenting in this paper, but they are not presented in a way that is easy to follow. I have summarized a list of things that can help improve this manuscript before it is published in AMT.

We thank the reviewer for the comments. In the following, we provide our point-to-point responses to these comments and list our revisions to the text in order to address these comments.

General comment:
In this paper, the authors rely a lot of statistics and try to interpret the difference (or bias) between the two data sets looking at all data as a population. It is one way of looking at things and to get a sense of the magnitude of this bias in all measurements as a whole. But for ambient measurements, sometimes it is also important to understand the time series of this deviation and to really understand the reason behind the bias. For example, does the bias is higher in summer than winter? Or is always consistent over the course of the sampling period? During the analysis, the authors discovered a significant number of samples that have no observable or detectable "OP". More investigation is needed to understand if this may impact the data differently for example over different seasons.

We understand and respect the perspective offered by the reviewer on a comprehensive way to carry out and interpret a comparison study like ours. It would be ideal if available data and resources could allow for a thorough investigation of various factors driving the differences, such as seasonality of chemistry, as mentioned by the reviewer, as well as source characteristics of the samples, etc. The samples analyzed here were collected during September to October of 2007 (Set 1) and May to September of 2017 (Set 2), which do not offer much information on the seasonal characteristics of inter-instrument differences. However, the dataset does cover a great variety of emission sources and meteorological conditions, given the wide spatial coverage of the CSN network, which makes it a comprehensive and well suitable statistical sample for us to achieve our primary goal of the study, i.e., to compare results obtained from three (Sunset vs. two DRI models) analyzers using the *same* analysis protocol, IMPROVE A with reflectance charring correction from the same sample. It should be mentioned that as valuable as such a comparison is for the community to make better use of these instruments and interpretations of the results, it has not yet been done for the CSN network until this study, mainly because of the large amount of work involved (e.g., a reanalysis of the archived samples). Given the comment by the reviewer, we have added a statement in the introduction section (Line 95-99) to emphasize the goal and contribution of this paper, i.e., statistical analysis of inter-instrument differences, that reads *"These samples, which were collected during September to October of 2007 (Set 1) and May to September of 2017 (Set 2), covered a great variety of emission sources and meteorological conditions, given the wide spatial coverage of the CSN network, ensuring statistically robust comparison among the three instrument models."*

The wide spatial coverage and hence variety of sources sampled are also mentioned in the revised experimental section 2.2.1 (Line 155-156): "*Both sets cover a variety of emission sources given the wide spatial coverage of CSN network.*"

We also add a statement in the conclusion section (Line 500-502) to recognize the potential value of a seasonal comparison: "*While data used in this study were primarily collected during the summer/fall season, future comparisons with data covering longer sampling period will paint a fuller picture of all seasons.*"

Last but not the least, it is our understanding that some of the reviewer's questions/concerns could be addressed by clarifying and emphasizing that all analyzers are using reflectance correction for charring. We have made such clarification throughout the revised manuscript where applicable.

In order to obtain the correct OC and EC, charred OC has to be taken into account. The Sunset instrument laser detector detects the transmittance whereas the DRI analyzer has the option to use either transmittance or reflectance. I could not find where the authors discuss about whether they are using transmittance or reflectance to determine charred OC (or OP).

The Sunset Model 5L analyzers used in our study are equipped with dual optical units, allowing for concurrent detection of both filter transmittance and reflectance (https://sunlab.com/wp-content/uploads/Lab-Instrument-brochure.pdf). All OC and EC data presented in this paper are corrected using reflectance. To make this clearer we have added the abovementioned detail about Sunset's dual optical units (Line 90) and clarified in multiple places where confusion might occur that the optical correction is by reflectance.

Also, in multiple places when OC and EC were discussed, are they both corrected for charred OC? Any comparison for OC and EC between the two data sets without accounting for charred OC is meaningless.

We acknowledge that the terms OC and EC are commonly referred to those carbon fractions with charring correction applied. In the original manuscript we have been careful with the different terms by using $OC_{sum}$ and $EC_{sum}$ when referring to the uncorrected OC and EC. It is our understanding that OC and EC subfractions are used widely by source apportionment studies as PM source markers [e.g., Liu et al., 2005], making us believe it is useful to emphasize the physical meanings of these subfractions to inform future use of these data. To avoid any confusion, in the revised paper we introduce these terms earlier in Sect 2.2.2 and use "$OC_{1+2+3+4}$" and "$EC_{1+2+3}$" instead of $OC_{sum}$ and $EC_{sum}$ to exclusively refer to the uncorrected OC and EC.

The authors introduce a new term, referred to as "SRD" or scaled relative difference. I understand this may have some statistical value, but this is not a typical common term that all readers will connect to when they read. This is particularly an issue when readers looking at various graphs generated from this value and try to understand or interpret its meaning. One way to improve this (if the authors insist of keeping this analysis) is to do a better job in explaining this term other than by defining it again. In other words, how readers should interpret this term for its magnitude. For example, when one express absolute difference between two numbers, readers understand the value of 0 means there is no bias. Or large positive or negative values means certain set is higher or lower. In comparison, SRD (because of the way it is defined), it does not easily translate.

Also, is this SRD term mostly used for interpreting the comparison between measurements with replicates? It is not clear from the heading of the sub sections to me.

We clarify that SRD is not a new term and actually has been used in many of our publications (Hyslop and White, 2009, Spada and Hyslop, 2018; Gorham et al., 2020). The merit of SRD is that it is the normalized difference between two measurements in %, and it considers measurement uncertainty from both measurements in a paired analysis. By using it for both within-analyzer and inter-analyzer comparisons, it is possible for one to easily compare the magnitude of the measurement uncertainty, determined from the replicate measurement, to the inter-analyzer difference. Considering the reviewer's comments, we do feel that it is worthwhile to further emphasize the premise and purpose of using SRD for the readers to better understand its meanings. We have expanded equation 5 to show that SRD = RD/$\sqrt{2}$ and revised the related text (Line 231-234) to emphasize its merit that reads "*The SRD, which equals relative difference (RD) divided by$\sqrt{2}$, is chosen over RD because it is the normalized relative difference between two measurements, accounting for the presence of equal and independent errors in both original and replicate measurements (Hyslop and White, 2009).*" We've also added a statement in the revised text (Line 269-270) that reads "*In both cases, a positive AD or SRD value occurs if the Sunset measurement is higher than the DRI measurement.*"

Specific comment:
p.7, line 275-285. It is observed that Sunset TC is generally agree with DRI TC, however, Sunset OC was found lower than DRI OC (and the opposite is true for EC). This is obviously due to the difference in OCEC split but is could also due to the fact that one instrument used transmittance to determine charred OC whereas the other one use reflectance. I couldn't find where this is discussed or whether the authors are using data to ensure the consistency (i.e., all corrected data are based on transmittance or reflectance).

As discussed in the response to the earlier comments, both analyzers are using reflectance charring correction.

p.9, line 339. Determining the correct OC and EC really requires the correction of OP. So what is the purpose of showing the "uncorrected" OC and EC when you know OP has to be taken into consideration?

The purpose is to point out the large difference in uncorrected OC and EC by the two analyzers using the same TOA protocol, IMPROVE A with reflectance correction, as this discrepancy likely arises from differences in instrument configuration and setting. We look at the difference between analyzers for both uncorrected and corrected OC and EC to examine and understand this difference.

Also, by definition, OP is subtracted from EC and added to OC to determine the final EC and OC. Therefore, in comparing the "uncorrected" and "corrected" OC, one should expect OC and EC be shifted in the same magnitude but in opposite direction. How come this does not seem to be the case in Fig. 4? Is this because it is plotted as a "relative" difference rather than absolute difference? Would absolute difference be more meaningful than the relative difference in this case?

The reviewer is correct about corrected OC and EC being shifted in the same magnitude, when the absolute difference is plotted as the y-axis. In Figure 4, the (scaled) relative difference is plotted instead of the absolute difference, thus the magnitude of change before and after charring correction is different for OC and EC. We choose to plot the relative difference over the absolute difference to show the different impact that the optical correction has on OC and EC because EC is a much smaller fraction of TC compared to OC.

The data in Fig. 4 are sorted according to the "mass loading percentage". How is this defined? Do the authors mean the amount of OC or EC relative to TC? So what is the advantage of plotting the data this way rather than sorting them by the absolute EC and OC mass?

The x-axis of Figure 4 is the sorted absolute OC and EC mass loadings, not their mass fractions in TC. The distribution of SRDs is plotted for each of the 20 mass loading bins (5th percentiles). The caption of Figure 4 and related text are revised to clarify this point.

p.9, line 348-350. The authors suggested for EC measurements with large SRD were samples with no instrumentally detected OP. What is considered "large"? Better to give a range or value or threshold. Does this represent bars with high "relative difference" in Fig 4?

As described in Sect 2.2.3, inter-model uncertainty derived from the replicate analysis is used as a benchmark for inter-model comparison. Both estimates are given in SRD, enabling a direct comparison. We have added the following statement to the revised text (Line 361-364): "*In investigating this anomaly, we found that EC SRDs were larger for samples with no instrumentally detected OP (i.e., OP = 0) by both Sunset and DRI-2015 analyzers (Figure 5a), with a median value of 20.9%, which far exceeds the inter-model uncertainty for EC determined from the replicate analysis (Table 3).*"

In addition, the authors said there are consider number of samples with no detectable OP, if I understand this correctly, no "OP" means the "corrected" OC (or EC) and "uncorrected" OC (or EC) are equal, correct? However, I don't ever see any evidence that there are any samples with "non-detectable OP" in Fig. 4. Fig. 4 shows that there are always a considerable amount of OP in all samples!

The reviewer's understanding is correct that corrected and uncorrected OC and EC are the same when OP is non-detectable. The reason the reviewer doesn't easily identify the OP=0 samples in Figure 4 is that the data are binned by mass loadings and what's shown in that figure is a distribution of samples in each bin. As Figure 5b shows, samples with OP = 0 are always a subset of samples in comparison in all EC mass loadings percentile bins, even at the highest EC mass loading percentile where OP = 0 samples comprise ~30% of the total samples in that bin.

p.9, line 355-359. The authors suggested that the EC bias almost solely originated from the thermal effect and I don't understand the argument supporting this statement. In Fig. 5, I only see the sub group of measurements with "no OP" (i.e., no thermal correction) has even higher relative difference than the group with considerable amount of OP. Or may be I don't understand how to interpret this graph. Plotting in relative difference could be one issue. If the authors' statement is true, I would expect for the sub-group of measurements with no OP should give you no bias between the two data set. Is this the case?

The sub-group of measurements with no OP has the largest bias in EC because of the large difference in the quantified OC subfractions (discussed in Section 3.1.2) by the two instrument models. To better guide the reader to interpret this figure, we have revised the related text and the new text (Line 371-377) reads: "*As shown in Figure 5c, for the 179 samples with no charring correction from both models, considerable correlation was found between the inter-model differences of EC and $OC_{1+2+3+4}$. This suggests that, in the absence of charring correction, much of the observed bias in EC between the two models is essentially coming from the inconsistency in the quantified OC subfractions by the two models. In contrast, samples with charring correction (i.e., OP > 0) showed little correlation between the inter-model biases of EC and $OC_{1+2+3+4}$.*"

p.9, line 360-386. This whole paragraph may be should not belong in main text. Probably is better to be included in supplementary information and then just refer to it in the main text. This paragraph is mostly to explain Fig. 6a, however, I would suggest the authors also include the detector signal, so the authors

would better understand how the OP peak compared to OC or EC and aid the interpretation of the thermograms. Only the laser response in Fig. 6 does not help much.

Thanks for the suggestion. We agree that FID signal would be a good addition to Figure 6 as the reviewer suggested. Thus, we have modified Figure 6 to include FID signal along with the laser response in the example thermograms from Sunset. However, after consideration, we believe that Figure 6 and the paragraphs discussing it are an important element of the main text. We have also modified Figure 6 to retain only the empirical cumulative plots (new Figure 7a and b) while removing the histograms, as the latter is in essence another angle looking at the statistical distributions but does not add much more information. The new Figure 6, Figure 7, and text of discussions are now more concise and more suitable to be shown in the main text.

Fig. 6b to e. What is the purpose of these? How does the different shapes in Fig. b and c are supposed to mean and what we should expect? I really don't think these graphs really aid much in terms of understanding what is going on regarding the "OP = 0" issue. The authors can however keep them in supplementary info if they want. What would really help is to actually include a completed thermograms including both laser response and detector response. Only that will allow the readers to understand how the samples evolve over the course of the analysis in a typical situation when OP=0 and OP>0 and may also include the case for blank as reference.

The purpose of Figure 6 (new Figure 7) is to offer a revelation of some unique characteristics of samples with OP=0, by closely examining the instrument signals and statistical populations of three types of samples, i.e., OP=0, OP>0 and blanks. Those OP=0 samples uniquely stand out with low initial and final laser readings, which are evident in both the example thermogram (Figure 6a) and the statistical distribution of laser readings (Figure 6b and 6c). After considerations of the last two comments on Figure 6, we have made two revisions to this figure, i.e., to add FID signals and to remove the histograms, which make it better emphasize these key characteristics of those OP=0 samples.